# Co-expression of CD39 and CD103 identifies tumor-reactive CD8 T cells in human solid tumors

Thomas Duhen[1], Rebekka Duhen[2], Ryan Montler[1], Jake Moses[1], Tarsem Moudgil[2], Noel F. de Miranda[3], Cheri P. Goodall[2], Tiffany C. Blair[1], Bernard A. Fox[2], Jason E. McDermott [4], Shu-Ching Chang[5], Gary Grunkemeier[5], Rom Leidner[2], Richard Bryan Bell[2] & Andrew D. Weinberg[1,2]

Identifying tumor antigen-specific T cells from cancer patients has important implications for immunotherapy diagnostics and therapeutics. Here, we show that $CD103^+CD39^+$ tumor-infiltrating CD8 T cells (CD8 TIL) are enriched for tumor-reactive cells both in primary and metastatic tumors. This CD8 TIL subset is found across six different malignancies and displays an exhausted tissue-resident memory phenotype. $CD103^+CD39^+$ CD8 TILs have a distinct T-cell receptor (TCR) repertoire, with T-cell clones expanded in the tumor but present at low frequencies in the periphery. $CD103^+CD39^+$ CD8 TILs also efficiently kill autologous tumor cells in a MHC-class I-dependent manner. Finally, higher frequencies of $CD103^+CD39^+$ CD8 TILs in patients with head and neck cancer are associated with better overall survival. Our data thus describe an approach for detecting tumor-reactive CD8 TILs that will help define mechanisms of existing immunotherapy treatments, and may lead to future adoptive T-cell cancer therapies.

[1] AgonOx, Inc., 4805 NE Glisan Street 2N35, Portland, OR 97213, USA. [2] Earle A. Chiles Research Institute, Providence Cancer Institute, 4805 NE Glisan Street 2N35, Portland, OR 97213, USA. [3] Department of Pathology, Leiden University Medical Center, P1-43, LUMC, Albinusdreef 2, 2333 Leiden, The Netherlands. [4] Pacific Northwest National Laboratory, Computational Biology and Bioinformatics Group, MSIN: J4-33, 902 Battelle Boulevard, PO Box 999, Richland, WA 99352, USA. [5] Medical Data Research Center, Providence Saint Joseph's Health, 9205 SW Barnes Road, Portland, OR 97225, USA. These authors contributed equally: Thomas Duhen, Rebekka Duhen, Andrew D. Weinberg. Correspondence and requests for materials should be addressed to T.D. (email: thomas.duhen@agonox.com) or to A.D.W. (email: andrew.weinberg@providence.org)

The immune system can recognize and destroy tumor cells through T-cell-mediated mechanisms. Hence, a variety of therapeutic approaches have focused on boosting and/or restoring T-cell function in cancer patients[1,2]. An effective immune response involves the concerted action of several different cell types among which CD8 T cells are key players that can specifically recognize and kill cancer cells via the release of cytotoxic molecules and cytokines[3]. A percentage of tumor-infiltrating CD8 T cells (CD8 TIL) recognize tumor-associated antigens, which include overexpressed self-antigens, as well as tumor-specific neoantigens, which arise as a consequence of tumor-specific mutations[4]. According to the current paradigm, tumor-specific CD8 T cells are primed in tumor-draining lymph nodes (LN) and then migrate via the blood to the tumor, where they exert their effector function. Previous work has shown that CD8 TILs represent a heterogeneous cell population comprising tumor-specific T cells as well as bystander T cells. Both tumor-specific and bystander T cells are recruited to the tumor site by the inflammation associated with tumor progression. However, it has proved difficult to easily identify cancer antigen-specific CD8 TILs within human tumors[5–8].

Recruitment and retention within the tumor requires T cells to express a defined set of chemokine receptors and integrins. Among the integrins, integrin αE, also known as CD103, is expressed on a subset of dendritic cells in the gut and a population of T cells found among peripheral tissues, known as tissue-resident memory T cells ($T_{RM}$)[9–11]. Several groups have shown that CD103 is also expressed on a subset of CD8 TILs in multiple solid human tumors[12–17] and it is known that TGF-β upregulates its expression[18].

More recently, the expression and function of CD39 and CD73 in human solid tumors has been of interest[19], especially with regard to treatments aimed at blocking their function[20]. CD39 is an ectonucleotidase expressed by B cells, innate cells, regulatory T cells as well as activated CD4 and CD8 T cells, which, in coordination with CD73 can result in local production of adenosine leading to an immunosuppressive environment. Furthermore, CD39 was identified as a marker for exhausted T cells in patients with chronic viral infections[21].

In this manuscript, we show that co-expression of CD39 and CD103 identifies a unique population of CD8 TILs found only within the tumor microenvironment. These cells, which have a $T_{RM}$ phenotype and express high levels of exhaustion markers, have a high frequency of tumor-reactive cells, have a distinct TCR repertoire and are capable of recognizing and killing autologous tumor cells. Finally, there is a greater overall survival (OS) in head and neck cancer patients that have a higher frequency of CD103+CD39+ CD8 TILs at time of surgery. These data provide an approach to identify tumor-reactive CD8 T cells and will have important ramifications for developing future therapeutic strategies.

## Results

### CD103 and CD39 identify tumor-resident CD8 T cells.
Recent work has shown that tumor-reactive CD8 T cells can be found within the CD103+ subset of TILs from patients with high-grade serous ovarian cancer (HGSC) and non-small cell lung cancer (NSCLC)[12,15]. However, repeated exposure to their cognate antigen can induce an exhausted state, ultimately impairing their capacity to control tumor growth[22–24]. Our preliminary data revealed that one of the top differentially expressed genes between CD103+ and CD103− CD8 TILs was *ENTPD1*, which encodes for the protein CD39 (data not shown), a marker for exhausted CD8 T cells in patients with chronic viral infections[21]. To further explore the CD103+ CD8 TIL subset, we examined the expression

of CD39, PD-1, CD127, and additional cell surface markers associated with activation/exhaustion in a patient with head and neck squamous cell carcinoma (HNSCC). Interestingly, CD103+ CD8 TILs expressed high levels of CD39, PD-1, and CD69 with a large fraction of cells co-expressing all three markers (Fig. 1a, b). CD39+PD-1+CD69+CD103+ CD8 TILs also expressed very low levels of CD127, thus displaying a chronic activation phenotype. Flow cytometric analyses of CD103 and CD39 on CD8 TILs revealed three main cell populations, which included CD103−CD39− double-negative (DN) T cells, CD103+CD39− single-positive (SP) T cells and CD103+CD39+ double-positive (DP) T cells, and a rare CD103−CD39+ T-cell population (SP39) (Supplementary Fig. 1a, b). DP CD8 TILs were detected at relatively high frequencies in tumor specimens obtained from patients with melanoma, lung cancer, HNSCC, ovarian cancer, and rectal cancer (Fig. 1c, d and Supplementary Fig. 2a). The percentage of DP CD8 TILs was lower in a subset of HNSCC patients as well as in most patients with microsatellite stable (MSS) colon cancer and colorectal liver metastasis (CRLM) (Fig. 1c, d). Interestingly, CD8 TILs from patients with DNA mismatch repair-deficient or microsatellite instability-high ($MSI^{high}$) colon cancer had a high frequency of DP CD8 T cells and this group has been associated with increased responsiveness to immunotherapy[25–28].

We next sought to determine whether the DP CD8 TILs could be found at primary and metastatic tumor sites in several HNSCC patients. Figure 1e shows results from a representative patient where DP CD8 T cells were found in both the primary tumor and metastatic LN; however, DP CD8 T cells were absent or present at very low frequency in the peripheral blood and an uninvolved LN. A similar distribution profile was found in most HNSCC patients analyzed (Fig. 1f). Collectively, these results indicate that CD39 expression on CD103+ CD8 T cells identifies a population of cells found predominantly in the tumor microenvironment.

### DP CD8 TILs are exhausted and display a $T_{RM}$ gene signature.
To gain a more in depth understanding of the CD8 TIL biology, the DN, SP, and DP CD8 T-cell subsets were sorted from the tumor and gene-expression profiles were assessed by microarray from three HNSCC and two ovarian tumors (Supplementary Fig. 2c). Volcano plot messenger RNA (mRNA) comparisons between the DP and DN CD8 T cells identified 219 transcripts that were differentially expressed between these two cell populations (Fig. 2a). Principal component analysis (PCA) and unsupervised hierarchical clustering revealed that the gene signature for DP and DN CD8 T cells were distinct, whereas the SP CD8 T cells displayed an intermediate gene signature, and those signatures segregated by CD8 T-cell subset rather than by patient or tumor type (Fig. 2b, c and Supplementary Table 1). Gene-set-enrichment analysis (GSEA) also revealed that gene transcripts associated with T-cell exhaustion such as *PDCD1* (PD-1), *CTL4* (CTLA-4), *HAVCR2* (TIM-3) were significantly enriched in DP CD8 as compared to DN CD8 and SP CD8 TILs while *CD28* expression was decreased (Fig. 2d, e and Supplementary Fig. 3). In contrast, DP CD8 TILs demonstrated lower expression of genes involved in T-cell recirculation, such as *KLF2*, *SELL* (CD62L), *S1PR1*, and as a result, showed a significant enrichment in the $T_{RM}$ gene signature when compared to DN CD8 and SP CD8 TILs (Fig. 2d, e and Supplementary Fig. 3).

To confirm the gene array results, we analyzed protein expression of several differentiation and activation markers by flow cytometry (Fig. 3a, b and Supplementary Fig. 2a). The DP CD8 TILs expressed higher levels of CD69 when compared to SP or DN CD8 TILs. CD69 is an activation molecule that is upregulated after T-cell stimulation and antagonizes sphingosine 1-phosphate receptor 1

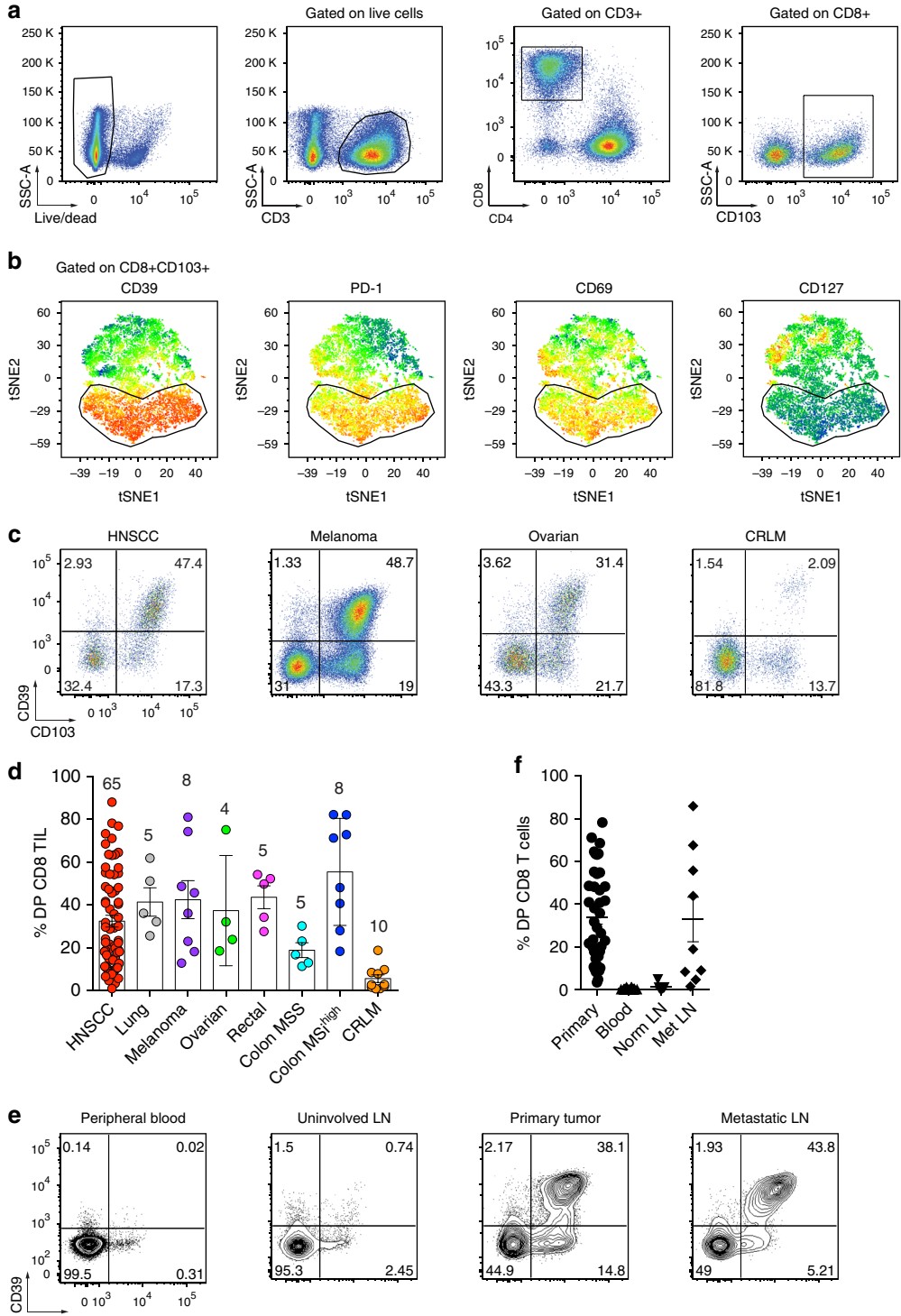

**Fig. 1** Tumor-infiltrating CD103[+] CD8 T cells coexpress the ectonucleotidase CD39. **a** Representative gating strategy for the flow cytometric analysis of tumor-infiltrating CD8 T cells (CD8 TIL) for a HNSCC patient. Numbers in plots indicate the percent cells in respective gates. **b** viSNE analysis of CD103[+] CD8 TILs as shown in **a**. The gate identifies the CD103[+] CD8 T cells with the highest expression of CD39. The gate is applied to all plots showing expression levels of PD-1, CD69, and CD127. **c** Flow cytometric analysis of CD8 TIL isolated from representative HNSCC, melanoma, ovarian cancer, and CRLM patients. Numbers in each quadrant indicate percent cells positive for CD39 and/or CD103 on CD3[+]CD8[+] T cells. **d** Summary of the frequency of CD39[+]CD103[+] (DP) CD8 TILs among patients with different solid malignancies. Shown are 65 HNSCC, 5 lung cancers, 8 melanomas, 4 ovarian cancers, 5 rectal cancers, 5 MSS colon cancers, 8 MSI[high] colon cancers, and 10 CRLM. **e** Flow cytometric analysis of the percentage of DP CD8 T cells in peripheral blood, uninvolved LN, primary tumor, and metastatic LN from a representative HNSCC patient. Numbers in each quadrant indicate percent cells positive for CD39 and/or CD103 on CD3[+]CD8[+] T cells. Data are representative of 41 HNSCC patients analyzed (7 patients for uninvolved LN and 9 patients for Met LN). Percentages are summarized in **f**. Small horizontal lines indicate mean ± SEM

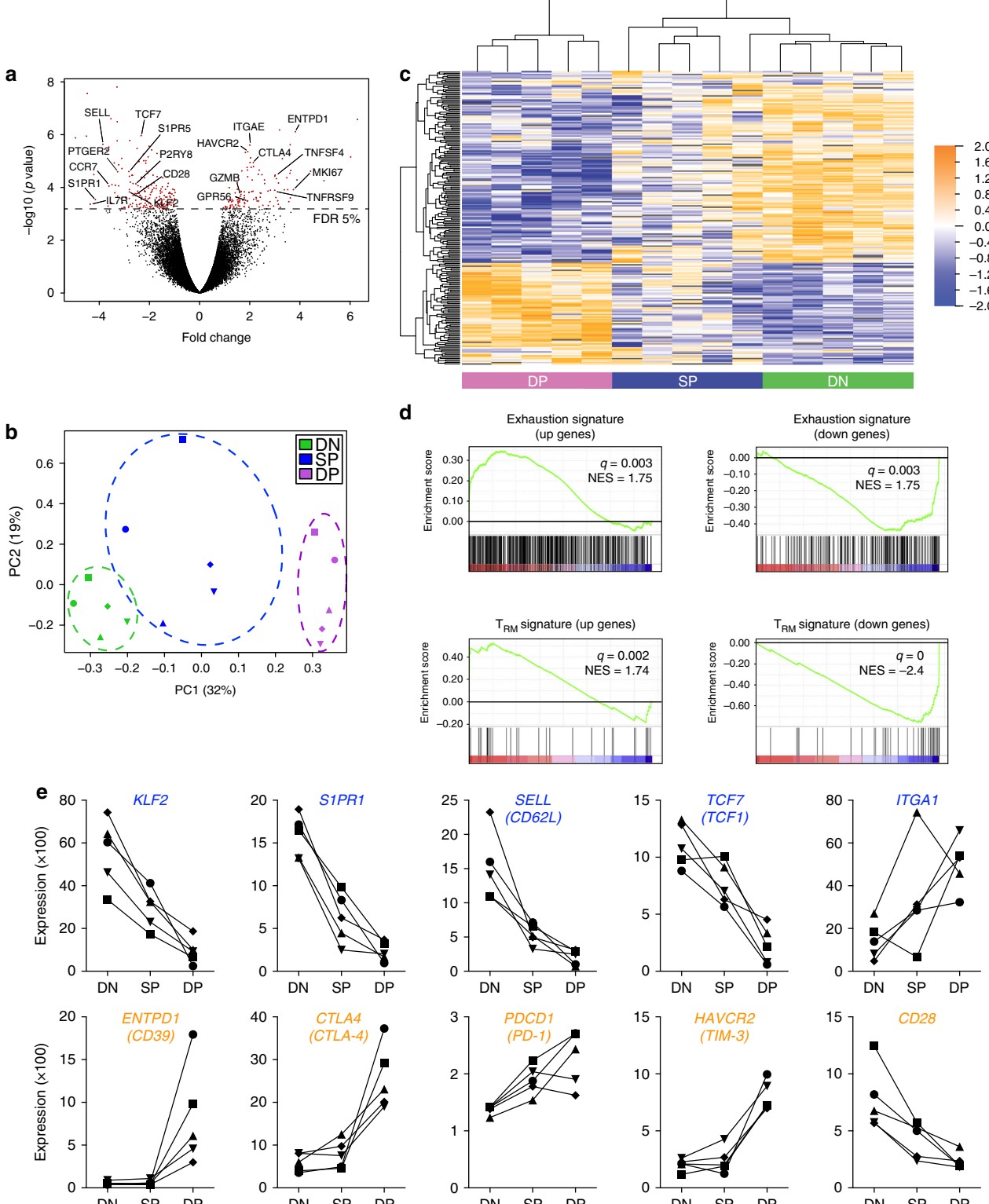

**Fig. 2** Gene-expression analysis of DP CD8 TILs reveals an exhausted profile and a gene signature reminiscent of T$_{RM}$ cells. **a** Volcano plot showing differential gene expression of the DP vs. DN CD8 T-cell subsets (log2-transformed). The dashed line delineates the differentially expressed genes (red dots) when using a false-discovery rate (FDR) <5%. **b** PCA of the 219 differentially expressed genes between the DP and DN CD8 TILs with a FDR <5% among all three subsets. Dots indicate samples of the three different populations from a total of five patients. PCA 1 and 2 represents the largest source of variation, combined accounting for 51%. Each symbol represents one patient. **c** Unsupervised clustering of the 219 differentially expressed genes between DP and DN CD8 on all three subsets of CD8 TIL from five patients. Blue color indicates downregulated genes, orange indicates upregulated genes. **d** GSEA of "T-cell exhaustion" and "T$_{RM}$" gene sets in the transcriptome of DP CD8 vs. that of DN CD8 TILs, presented as the normalized enrichment score (NES). **e** Examples of differentially expressed genes between DP CD8 and DN CD8 TILs present in the "T$_{RM}$" (blue) or the "T-cell exhaustion" (orange) gene sets. Each symbol/line represents one patient, which is connected by a line

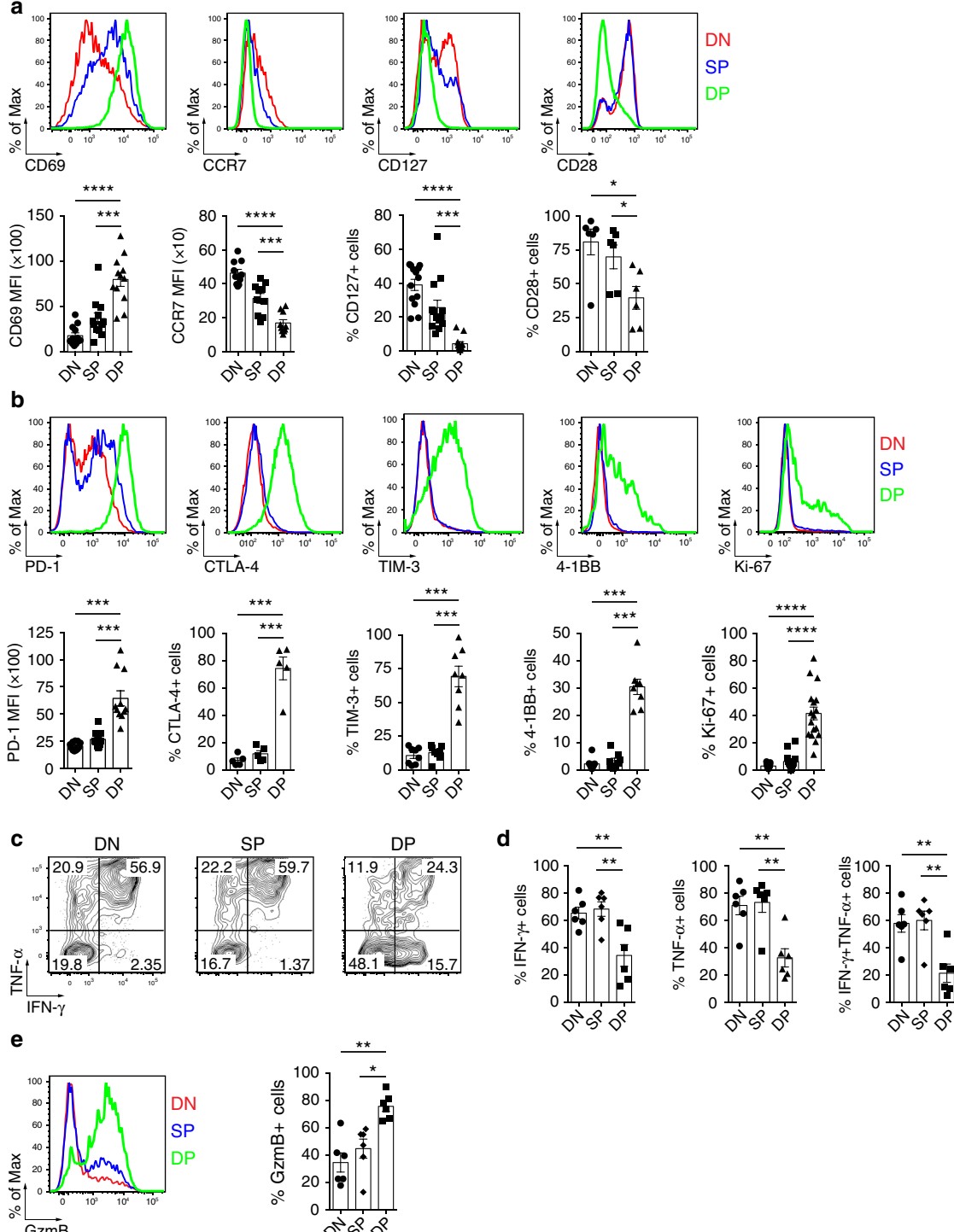

**Fig. 3** Phenotypic and functional properties of DP, SP and DN CD8 TILs. **a** Ex vivo phenotypic analysis of the expression of surface proteins CD69, CCR7, CD127, and CD28, delineating a T$_{RM}$ phenotype, on the three CD8 TIL subsets (top, one representative patient shown, colors match key). Summary of the frequency/expression level of the above-mentioned markers among several cancer patients (n = 12 for CD69; n = 10 for CCR7; n = 13 for CD127; and n = 6 for CD28, bottom). Each dot represents one patient. **b** Ex vivo phenotypic analysis of the expression of surface and intracellular proteins PD-1, CTLA-4, TIM-3, 4-1BB, and Ki-67, delineating an activated/chronically stimulated phenotype, on the three CD8 TIL subsets (top, one representative patient shown, colors match key). Summary of the frequency of the above-mentioned markers among several cancer patients (n = 11 for PD-1; n = 5 for CTLA-4; n = 8 for TIM-3; n = 8 for 4-1BB; and n = 17 for Ki-67, bottom). Each dot represents one patient. **c** Representative flow cytometric analysis of IFN-γ and TNF-α production by the three CD8 TIL subsets stimulated for 4 h with Phorbol 12-Myristate 13-Acetate (PMA)/ionomycin. Numbers in each quadrant indicate percent cells positive for IFN-γ and/or TNF-α for each subset. **d** Frequency of IFN-γ, TNF-α, IFN-γ/TNF-α double-positive cells by each CD8 TIL subsets. Data are from 6 HNSCC patients. **e** Representative flow cytometric analysis of granzyme B production by CD8 TIL subsets (left, colors match key) and summary of six different HNSCC patients (right). Small horizontal lines indicate mean ± SEM; *p < 0.05; **p < 0.01; ***p < 0.001; ****p < 0.0001; p-values were determined by one-way analysis of variance with Tukey's post hoc testing (**a**, **b**, **d**, **e**)

(S1PR1)-mediated egress of T cells from tissue, thereby playing a role in tissue retention[29]. The DP CD8 TILs also exhibited lower expression of CCR7, CD127, and CD28, indicative of an effector memory phenotype[30]. Interestingly, while a fraction of all CD8 TIL sub-populations expressed PD-1 (Supplementary Fig. 4), the level of PD-1 expression was highest on the DP CD8 TILs. In addition, CTLA-4 and TIM-3 were almost exclusively expressed within the DP CD8 T-cell population. The DP CD8 TILs also displayed increased frequencies of cells expressing 4-1BB and Ki-67, suggesting that these cells had recently encountered their cognate antigen and were proliferating in the tumor.

Effector function of the DN, SP and DP CD8 T cells was assessed by analyzing cytokine production and granzyme B expression. We found that fewer DP CD8 TILs produced interferon (IFN)-γ and/or tumor necrosis factor (TNF)-α than did the DN and SP CD8 TILs after Phorbol 12-Myristate 13-Acetate (PMA)/ionomycin treatment (Fig. 3c, d). Interestingly, the DP CD8 TILs had an increased cytotoxic potential as demonstrated by a significantly higher frequency of granzyme B-positive cells (Fig. 3e).

**TGF-β and TCR stimulation upregulate CD39 and CD103.** Given that DP CD8 T cells were found only at sites where tumor cells were present, we wanted to explore the mechanism for CD39 and CD103 upregulation. It is well established that TGF-β drives expression of CD103 on T cells[18]. TGF-β is often produced within the tumor microenvironment and may play a major role in cancer progression in part by suppressing T-cell immunity[31,32]. CD39 expression is found on exhausted CD8 T cells in mice and humans[21,33], suggesting that chronic TCR stimulation may upregulate CD39. Therefore, we analyzed the kinetics of CD103 and CD39 upregulation on naive CD8 T cells from several healthy donors stimulated via the TCR in the presence or absence of

TGF-β (Fig. 4 and Supplementary Fig. 5). TGF-β greatly increased CD103 expression 3 days following TCR stimulation (75 ± 16.8%). CD39 expression, which was detected 3 days after TCR activation, peaked at day 9. Maximum CD39 expression required prolonged TCR stimulation as it was only partially upregulated when the TCR stimulation was removed after 24 h. The lack of CD39 upregulation was not due to sub-optimal TCR stimulation as the expression of other activation markers such as PD-1 was induced on T cells stimulated for only 24 h (Fig. 4). Taken together, these data show that prolonged TCR stimulation in the presence of TGF-β is necessary for the maximum co-expression of CD103 and CD39 on CD8 T cells.

**TCR repertoire analysis suggests clonal expansion of DP CD8.** Thus far, our data suggest that DP CD8 TILs may be enriched for cells recognizing cognate antigens within the tumor. If this assumption is correct, we might observe selective expansion of dominant TCR clonotypes within the DP CD8 TILs. Sequencing of the CDR3 region of the *TRB* genes revealed that DP CD8 TILs were indeed more oligoclonal compared to memory CD8 T cells in the peripheral blood and to DN and SP CD8 TILs (Fig. 5a). The 30 most frequent clonotypes were evaluated in a Human Papillomavirus (HPV)-positive HNSCC, a HPV-negative HNSCC, an ovarian tumor, and a melanoma tumor, and they accounted for 56, 62, 66 and 86%, respectively of the DP CD8 TILs; however, the top 30 clonotypes within the DN CD8 TILs only represented 26, 23, 38, and 35% in the same patients. The 30 most frequent clonotypes in the DP CD8 TILs were far less frequent in the DN or the SP CD8 TILs, and represented <2.5% of the DN CD8 TIL repertoire within these four patients (Supplementary Fig. 6a). Consequently, very few TCR clonotypes were shared between the DP CD8 TILs and the other CD8 TIL subsets (Fig. 5b). In contrast, a greater number of TCR clonotypes were shared between DN CD8 TILs and SP CD8 TILs. The majority of TCR clonotypes present in DN CD8 TILs were also shared with memory CD8 T cells within an uninvolved LN and in peripheral blood ($R^2 = 0.7710$ and $0.2022$, respectively) (Fig. 5c). These data suggest that the DN CD8 TILs can exit the tissue into the periphery as would be predicted by the lack of a resident memory signature (Fig. 2d). In contrast, the clonotypes present within the DP CD8 TIL repertoire were detected at very low frequency in the aforementioned peripheral tissues ($R^2 = 0.0026$ and $R^2 = 0.0038$, respectively) (Fig. 5c), suggesting that these cells were preferentially retained within the tumor. Calculation of the Morisita index, an abundance-based similarity index that determines the overlap between two populations, further supports these results showing consistent findings across six patient samples (Fig. 5d). Similar results were also obtained when calculating the Jaccard index with these samples (Supplementary Fig. 6b). Of note, we also observed a strong overlap of the DP CD8 TIL repertoire between the primary tumor and metastatic LNs in two HNSCC patient samples (Supplementary Fig. 6c). Collectively, these results support antigen-driven stimulation of the DP CD8 T cells at the tumor site, which results in local activation and expansion of these cells.

**DP CD8 TILs recognize and kill tumor cells.** To determine whether the DP CD8 TILs were enriched for cells that recognize and kill autologous tumor cells, we used tumor cell lines generated from four melanoma patients and two HNSCC patients in T cell/tumor co-culture experiments. CD8 T cells were sorted from tumor digests based on the expression of CD103 and CD39 and expanded in vitro (Supplementary Fig. 2c). Following expansion, the DN, SP, and DP CD8 TILs were screened for reactivity to autologous tumor cell lines by measuring upregulation of the

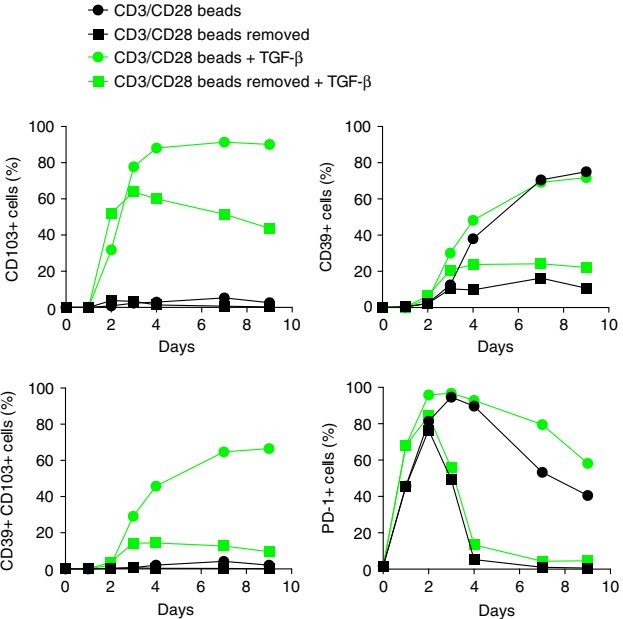

**Fig. 4** TGF-β and sustained TCR stimulation are required for CD103 and CD39 upregulation on CD8 T cells. Kinetics of CD39, CD103, and PD-1 expression on sorted naive CD8 T cells from peripheral blood after in vitro activation. Cells were stimulated with CD3/CD28-coated beads at a bead: T-cell ratio of 1:2 in the presence or absence of TGF-β1 (2 ng/ml), and expression of CD39, CD103, and PD-1 was analyzed by flow cytometry at the indicated time points. Data are from one representative healthy donor

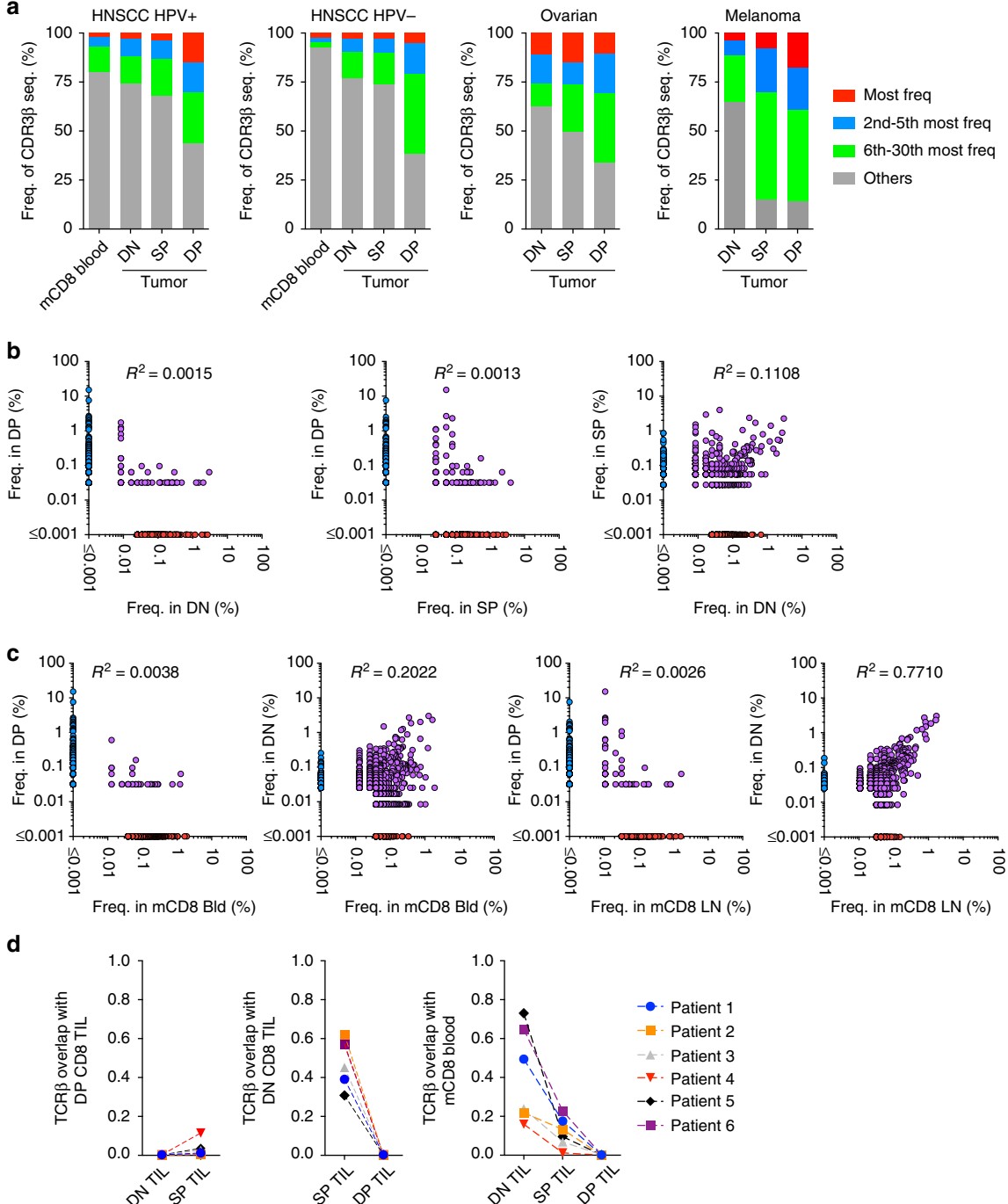

**Fig. 5** DP CD8 T cells are highly clonal and share little overlap with other CD8 TILs or peripheral CD8 T cells. **a** Diversity of the TCRβ repertoire within blood memory CD8 T cells, DN, SP, and DP CD8 TILs. The frequencies of the most frequent clonotype, the 2nd to 5th most frequent, the 6th to 30th most frequent, and the rest of the clonotypes (others) are shown for a HPV-positive HNSCC patient, a HPV-negative HNSCC patient, an ovarian cancer patient, and a melanoma patient. **b** The 500 most frequent CD8 TIL clonotypes for each subset are plotted based on their frequency in DN, SP, and DP CD8 TILs. Each dot represents a distinct TCR clonotype. Dots on the axis indicate the clonotypes detected within a single repertoire; purple dots indicate clonotypes shared between two CD8 T-cell populations. **c** Same analysis as in **b** comparing the frequency of the top 500 clones in memory CD8 T cells in peripheral blood and uninvolved LN to the frequency in DN and DP CD8 TIL subsets. **d** Similarity between the TCR repertoires of CD8 T-cell subsets was measured using the Morisita–Horn index on six cancer patients connected with a dashed line

activation marker 4-1BB and IFN-γ secretion. In all six patients, the DP CD8 TILs exhibited greater tumor reactivity when compared to the DN or SP CD8 TILs (Fig. 6a and Supplementary Fig. 7a). The frequency of tumor-reactive CD8 TILs within the DP population was 87% at the highest tumor-to-T-cell ratio for patient 1, and a mean of 51% tumor-reactive cells was found over

the six patient samples analyzed (ranging from 13.2 to 87.9%). The DP CD8 TILs specifically recognized autologous tumor cells in an major histocompatibility complex (MHC)-class I-dependent manner, as little to no reactivity was observed following Major Histocompatibility Complex (MHC)-class I blockade or when T cells were co-cultured with an allogeneic

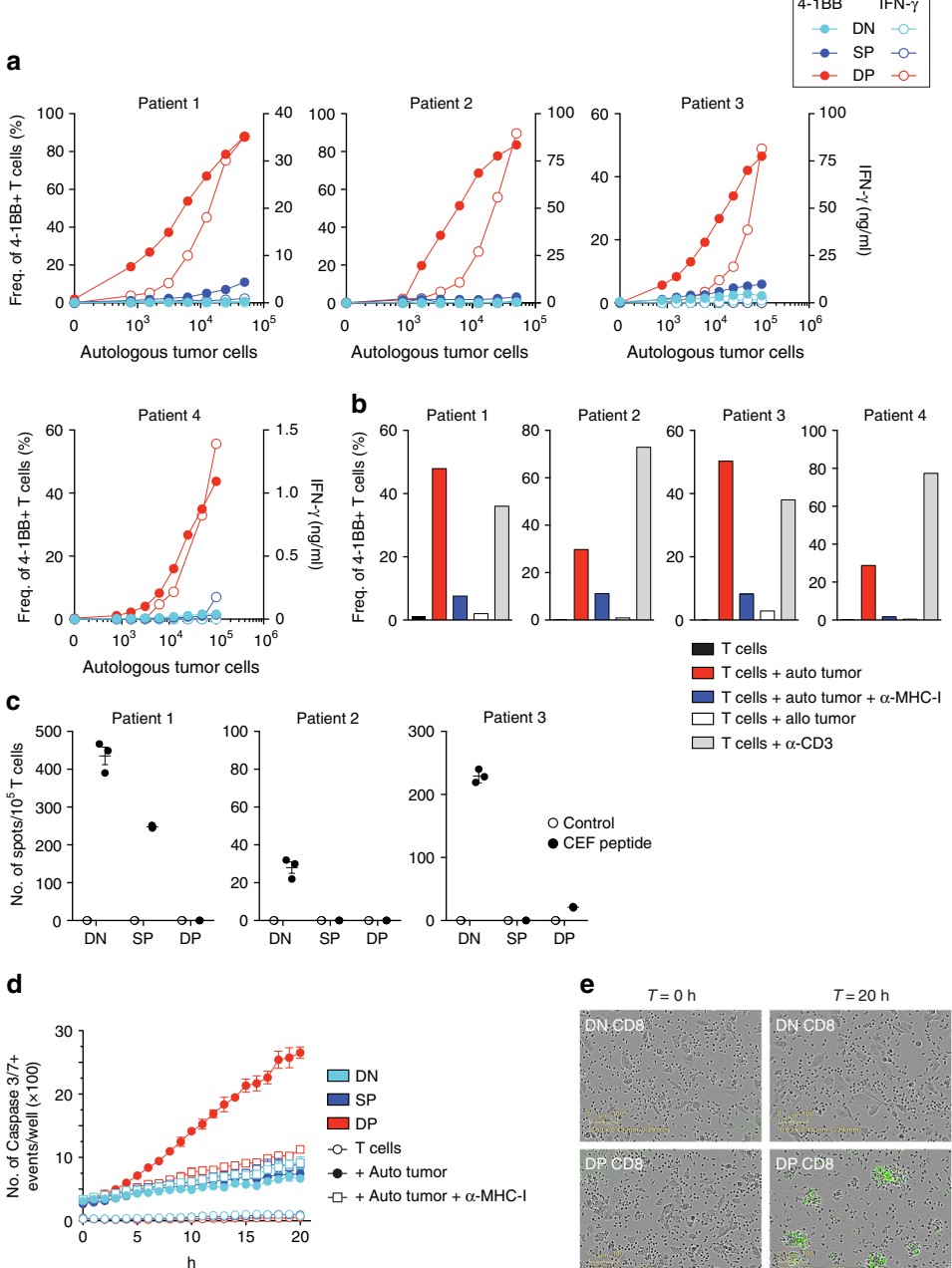

**Fig. 6** DP CD8 TILs recognize and kill autologous tumor cells. DN, SP, and DP CD8 TILs were sorted from tumor digest and expanded in vitro. **a** Expanded CD8 TILs were tested for tumor reactivity by cultivating them for 20 h with increasing numbers of autologous tumor cells, and tumor recognition was assessed by measuring the frequency of 4-1BB expression (filled symbols) and IFN-γ secretion (open symbols). Results are shown for four melanoma patients. **b** Reactivity of DP CD8 TILs was confirmed by culture with autologous tumor cells with and without MHC-I blocking antibody, allogeneic tumor cells, and plate-bound anti-CD3. The upregulation of 4-1BB after 20 h is shown for four melanoma patients. **c** The reactivity of expanded DN, SP, and DP CD8 TILs against a pool of 32 peptides derived from EBV, CMV, and influenza virus was assessed by measuring IFN-γ secretion by ELISPOT. Experiments were performed in triplicate and results are shown for three melanoma patients. All TIL sub-populations showed similar levels of IFN-γ when stimulated with anti-CD3. **d** Expanded DN, SP, and DP CD8 TILs were cultured with and without autologous tumor cells, in presence and absence of MHC-I blocking antibody. Caspase 3/7+ events (green fluorescence) were monitored every hour with the Incucyte live-cell imaging system over a 20 h period and analyzed as described in the Methods section. Small horizontal lines indicate the mean ± SEM. **e** Representative images for DN and DP CD8 T cells taken at the beginning (T = 0) and at the end of the co-culture with autologous tumor cells (T = 20 h). Green fluorescence indicates caspase 3/7 activity. Results shown in **d** and **e** are representative of two patients

tumor cell line (Fig. 6b and Supplementary Fig. 7b). The sorted DP CD8 TILs were not enriched for viral-specific CD8 T cells as measured by IFN-γ secretion after incubation with a peptide pool derived from Epstein-Barr virus (EBV), Cytomegalovirus (CMV) and influenza virus (Fig. 6c and Supplementary Fig. 8) (no viral

reactivity was detected in patient 4). To address whether the tumor-reactive DP CD8 TILs could directly kill autologous tumor cells, we co-cultured the expanded T-cell subsets with autologous tumor cell lines and monitored tumor killing. In accord with 4-1BB upregulation, the DP CD8 T cells were able to kill autologous

tumor cells in an MHC-class I-dependent manner as illustrated by the increasing number of caspase 3/7+ events (Fig. 6d, e). In contrast, little to no tumor cell killing was observed in the DN or SP CD8 TIL populations (Fig. 6d, e). Taken together, our data show that CD39 and CD103 co-expression strongly enriches for CD8 T cells that recognize and kill tumor cells.

**Frequency of DP CD8 in tumors correlates with increased OS.** Since the DP CD8 TILs were enriched for tumor-reactive T cells, we explored the possible relationship between the frequency of DP CD8 TILs and OS in a small cohort of HNSCC patients ($n$ = 62). Following surgery, the frequency of DP CD8 T cells among total CD8 TILs was determined within primary tumors by flow cytometry and patients then received standard of care treatment (s). Patients were segregated based on the mean frequency of the DP CD8 T cells within this cohort and their characteristics are detailed in Supplementary Table 2. Our analysis revealed that there was a significant difference in stage between DP-low and DP-high groups. Patients with lower frequency of DP CD8 T cells were more likely to have a higher pathologic stage (86% vs. 58%, $p$-value = 0.02), but there were no significant differences with respect to the other characteristics tested. Univariate survival analysis using log-rank test showed that patients whose tumors had a higher percentage of DP CD8 TILs at the time of surgery correlated with better OS (3-year survival: 89% vs. 56%; $p$-value = 0.0191) (Fig. 7a and Supplementary Table 5). This association was more significant in the HPV-negative subgroup (3-year survival: 86% vs. 26%; $p$-value = 0.0119) (Fig. 7b and Supplementary Table 6). In contrast, an increase in the frequency of SP CD8 TILs did not correlate with increased OS (Fig. 7c, d). There was no significant association between stage and OS (Supplementary table 3), probably due to small sample size and small number of events. To account for confounding variables when evaluating the association of DP CD8 TILs with OS, we took HPV status as well as stage, age, gender, and smoking status into

consideration in the multivariable analysis, though they did not show significant association with survival in univariate analysis. The results showed that, after controlling for those variables, higher percentage of DP CD8 TILs at the time of surgery was associated with better OS (hazard ratio (HR) = 0.23; 95% CI = 0.04–0.96; $p$-value = 0.04; Supplementary Table 3), and this association remained more significant in the HPV-negative subgroup (HR = 0.07; 95% CI = 0.003–0.54; $p$-value = 0.01; Supplementary Table 4). Specifically, taking DP CD8 as a continuous variable with log(2) transformed (i.e., log(DP CD8, 2)) in the HPV-negative subgroup showed that controlling for the same variables, the hazard of death significantly decreased 45% with two-fold increase of DP CD8 (HR (95% CI) = 0.55 (0.32, 0.95), $p$-value = 0.03). To confirm our data, we interrogated the TCGA data sets for HNSCC, lung adenocarcinoma, and lung squamous cell carcinoma, comparing patients with high expression of *ENTPD1* (CD39) and *ITGAE* (CD103) transcripts to patients with lower expression of those genes. Using this approach, we showed that patients in the CD39hiCD103hi group demonstrated a better OS at 3 years than patients from the CD39loCD103lo group for all three cancer types (Supplementary Fig. 9). Importantly, using both genes for the analysis was consistently better at predicting survival than CD103 alone.

Collectively, we show that the frequency of DP CD8 T cells in primary tumors correlates with increased OS in a cohort of HNSCC patients and within the TCGA data sets for HNSCC and lung cancer patients.

## Discussion

In this study, we show that within the CD8 TILs, the CD103+CD39+ cells are enriched for tumor-reactive T cells in human solid tumors. These cells have an exhausted $T_{RM}$ phenotype, a TCR repertoire that is distinct from other CD8 TIL subsets, and can kill autologous tumor cells in a MHC-restricted manner. Importantly, we found that increased frequencies of the

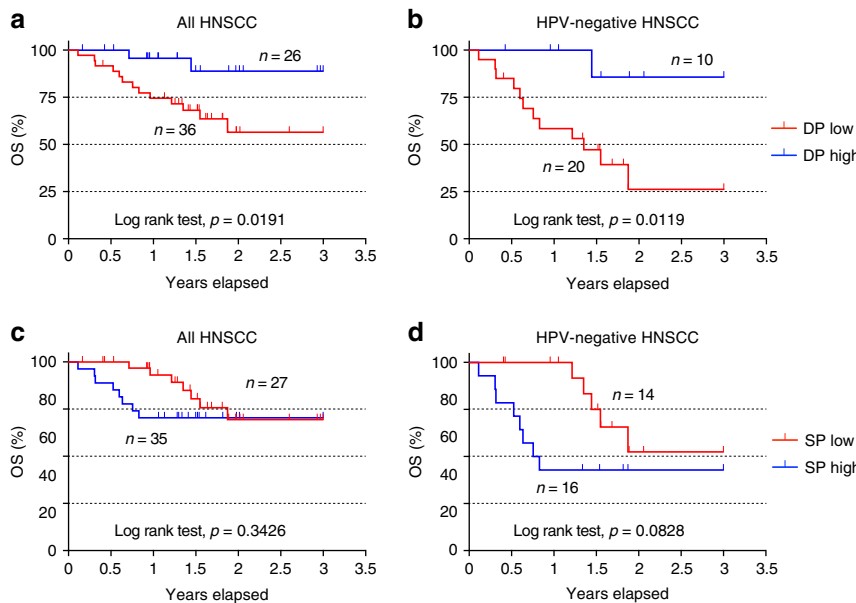

**Fig. 7** High frequency of DP CD8 TILs is associated with better overall survival in a cohort of HNSCC patients. Following surgery, the frequency of DP (**a**, **b**) and SP (**c**, **d**) CD8 T cells among total CD8 TILs was determined within the primary tumor by flow cytometry. **a** HNSCC patients ($n$ = 62) were segregated based on the frequency of the DP CD8 T cells, with DP high and DP low groups stratified above or below the mean frequency (32.9%) followed by Kaplan–Meier analysis. **b** Similar analysis performed on the HPV-negative HNSCC patients ($n$ = 30). **c** HNSCC patients ($n$ = 62) were segregated based on the frequency of the SP CD8 T cells, with SP high and SP low groups stratified above or below the mean frequency (18%) followed by Kaplan–Meier analysis. **d** Similar analysis performed on the HPV-negative HNSCC patients ($n$ = 30). The log-rank test was used to compare curves and $p$-values <0.05 were considered significant

DP CD8 TILs correlate with a significantly reduced risk of death for patients with HPV-negative HNSCC, suggesting an important role for tumor antigen-specific CD8 TILs in limiting tumor growth in these patients.

It has previously been suggested by several groups that CD103[+] CD8 TILs contained tumor-reactive CD8 T cells in HGSC and NSCLC[12,15]. Moreover, similarly to our work, the study by Ganesan et al.[34] recently showed that CD103[+] CD8 TILs displayed a $T_{RM}$ gene signature and were highly activated, supporting a common gene signature for CD103[+] CD8 TILs in multiple solid tumors. However, our study shows that gating on the CD39[+] cells within the CD103[+] CD8 TILs population is essential to enrich for tumor-reactive CD8 T cells, as the SP CD8 TILs showed little reactivity to autologous tumor cells (Fig. 6a and Supplementary Fig. 7a). In addition, the analysis of the TCR repertoire and gene signatures further indicated that this subset is a unique CD8 TIL sub-population.

The DP CD8 TILs were present at high frequencies within primary tumors and tumor-infiltrating LNs; however, there was little to no expression of these two markers on peripheral CD8 T cells (blood or uninvolved LN; Fig. 1e, f) within tumor-bearing hosts. Several different tumor types were assessed and the highest percentage of DP CD8 TILs were found in melanoma and MSI-high colon cancer, both tumors with high mutational burden that show the highest response rates to immunotherapy[26,27], while intermediate percentages were found in HNSCC, lung, and ovarian cancer patients (all known to respond to PD-1 blockade but to a lesser extent than melanoma and MSI[high] colon cancer) (Fig. 1d). Strikingly, we found fewer DP CD8 TILs in primary MSS colon cancer and CRLM, and these patients tend not to respond well to immunotherapy. Our data suggest that MSS colon cancer patients have a lower frequency of tumor-reactive TILs compared to other malignancies; however, being able to identify the tumor-reactive TILs in these patients may allow for their expansion and future immunotherapy treatment options.

Recent advances in cancer immunotherapy treatment have shown that adoptive T-cell therapy was effective in some tumor types, particularly in stage IV melanoma patients[1,35–38]. However, efficacy of this treatment strategy is tightly linked to the identification and in vitro expansion of tumor-reactive T cells. Currently, PD-1 is a marker used to identify antigen-reactive T cells among TILs and/or peripheral blood[5,39–41]. Here we show that PD-1[+] cells were detected in all CD8 TIL sub-populations (Supplementary Fig. 4); however, only the DP CD8 TIL subset was enriched for tumor recognition and immune specific tumor killing (Fig. 6a, d). Therefore, using CD39 and CD103 to enrich for tumor-reactive CD8 T cells prior to TIL expansion may be a way to increase the clinical success of adoptive immunotherapy. Also, identifying TCR sequences within the DP CD8 TILs that specifically recognize tumor antigens may aid in TCR engineering approaches that would lead to personalized tumor therapies in patients that have failed approved treatments.

The activated phenotype of the DP CD8 TILs and their skewed TCR repertoire strongly suggest that these cells are recognizing their cognate antigen within the tumor microenvironment. However, in this study, the CD8 TILs were isolated from progressively growing tumors, highlighting the fact that tumors can develop immune escape mechanisms that ultimately prevent the tumor-reactive CD8 TILs from controlling tumor growth. A potential explanation is chronic TCR stimulation within the tumor resulting in increased expression of activation/exhaustion markers, some of which participate in the downregulation of immune function (e.g., PD-1, CTLA-4 and TIM-3). Sustained expression of these checkpoints molecules can have detrimental effects on the immune response resulting in tumor outgrowth[42]. Therefore, using checkpoint inhibitors could release inhibition on

the tumor-reactive DP CD8 TILs allowing them to fully exert their effector function. To address this hypothesis, it would be of interest to examine patients that respond to anti-PD-1 and anti-CTLA-4 therapies for changes in the frequency and/or activation status of the DP CD8 TILs. CD39 is another potential immune inhibitory pathway, which is highly expressed on the DP CD8 TILs. CD39 is an ectonucleotidase that hydrolyzes extracellular ATP and ADP into AMP, which is then processed into adenosine by CD73, an ecto-5′-nucleotidase expressed by many tumor cell types[19]. Adenosine is a potent immunoregulator and, when binding to A2A receptors on T lymphocytes, it enhances the accumulation of intracellular cAMP, thereby dampening T-cell activation[43]. Interfering with the adenosine pathway via blockade of the CD39/CD73 axis or adenosine receptors on T cells within the tumor microenvironment might be another promising therapeutic strategy to increase the function/potency of the DP CD8 TIL and this pathway is currently being tested in cancer-bearing hosts[20].

Based on the aforementioned expression of inhibitory/checkpoint receptors, CD8 T cells have been classified and termed exhausted, anergic and/or dysfunctional. A recent study has reported that T-cell dysfunction is epigenetically imprinted and that two dysfunctional states exist, a "plastic" state from which T cells can be rescued and a "fixed" state, in which T cells are resistant to reprogramming[44]. Based on their data and the ex vivo phenotype of DP CD8 TILs defined within this manuscript, one could conclude that the majority of the CD8 TILs are dysfunctional. However, our cohort of HNSCC patients indicates that a higher frequency of DP CD8 TILs is associated with a lower risk of death, suggesting that tumor antigen-specific CD8 TILs may play an active role in the control of tumor growth. Moreover, we have demonstrated that the DP CD8 TILs can kill autologous tumor cells efficiently, suggesting that these T cells are able to regain function after in vitro expansion. Hence, the plasticity and heterogeneity of the DP CD8 TILs will be explored in future studies to further elucidate the functional role of these cells in anti-tumor immunity.

In conclusion, we propose a model where CD8 T cells are primed by dendritic cells presenting tumor antigens within the tumor-draining LN, then migrate to the tumor where they recognize their cognate antigen and clonally expand. The consequence of this recurrent TCR activation in a TGF-β-rich environment is the upregulation of CD39 and CD103 on CD8 TILs. Activation of these cells also leads to the downregulation of proteins important for T-cell recirculation resulting in the retention of DP CD8 TILs within the tumor. Finally, repetitive TCR stimulation of the DP CD8 TILs leads to impaired effector function, immune escape, and ultimately tumor progression. We believe that this work leads to a deeper understanding of human CD8 TIL biology, and using CD39 and CD103 to identify tumor-reactive CD8 TILs in solid tumors will lead to important mechanistic insights when evaluating the efficacy of immunotherapeutic treatments.

## Methods

**Healthy donor and patient samples**. Peripheral blood, uninvolved LNs, metastatic LNs, and tumor samples were obtained from individuals with HNSCC, melanoma, colon cancer, rectal cancer, lung cancer, CRLM, and ovarian cancer. All subjects signed written informed consent approved by the Providence Portland Medical Center Institutional Review Board (IRB protocol no. 06-108A). At the time of sample collection, patients were not undergoing therapy. Previously, patients had undergone a wide range of therapies, including chemotherapy, radiotherapy, surgery and immunotherapy, or none of the above. Peripheral blood mononuclear cells (PBMCs) were purified from whole blood over Ficoll-Paque PLUS (GE Healthcare) gradient and cryopreserved prior to analysis. Tumor specimens were prepared as follows: under sterile conditions, tumors were cut into small pieces and digested in RPMI-1640 supplemented with hyaluronidase at 0.5 mg/ml, collagenase at 1 mg/ml (both Sigma-Aldrich), DNase at 30 U/ml (Roche) as well as human

serum albumin (MP Biomedicals) at 1.5% final concentration. Cells were digested for 1 h at room temperature under agitation with a magnetic stir bar. Cell suspensions were filtered through a 70 μm filter. Tumor-infiltrating lymphocytes were enriched as described above by Ficoll-Paque PLUS density centrifugation. Tumor single-cell suspensions were cryopreserved until further analysis.

**Antibodies and flow cytometry.** The following fluorescent-labeled antibodies were used: fluorescein isothiocyanate (FITC), allophycocyanin (APC)-Cy7, and PE/Dazzle 594 anti-CD3 (UCHT1; 1:50—#300440, 1:100—#300426 and 1:100—#300450, respectively), BV785 anti-CD4 (OKT-4; 1:100—#317442), BV510 anti-CD8 (RPA-T8; 1:100—#301048), BV650 anti-CD25 (BC96; 1:100—#302634), PE/Dazzle 594 anti-CD28 (CD28.2; 1:50—#302941), APC-efluor780 anti-CD45RA (HI100; 1:100—#47-0458-42), BV605 anti-CD69 (FN50; 1:50—#310938), BV421 anti-CD127 (A019D5; 1:50—#351310), AlexaFluor (AF)-647 anti-PD-1 (EH12.2H7; 1:50—#329910), APC anti-CTLA-4 (BNI3; 1:25—#369611), PE anti-4-1BB (4B4-1; 1:40—#309804), PE/Dazzle 594 and BV421 anti-CCR7 (G043H7; 1:50—#353236 and #353208, respectively), AF700 anti-granzyme B (GB11; 1:100—#560213), BV605 anti-IFN-γ (4S.B3; 1:50—#502536), APC anti-TNF-α (Mab11; 1:200—#17-7349-82) (all from Biolegend); BV421 anti-CD27 (M-T271; 1:50—#562513), FITC anti-CD127 (HIL-7R-M21; 1:10—#560549), PE-Cy7 anti-PD-1 (EH12.1; 1:50—#561272), AF 488 anti-Ki-67 (B56; 1:100—#561165) (all from BD Biosciences); PE anti-CD3 (UCHT1; 1:100—#300408), PE anti-CD28 (CD28.2; 1:50—#12-0289-42), APC and PE-Cy7 anti-CD39 (eBioA1; 1:100—#17-0399-42 and #25-0399-42, respectively), PE and PerCP-efluor710 anti-CD103 (B-Ly7 and Ber-ACT8; 1:100—#12-1038-42 and 1:50—#46-1037-42, respectively) (all from eBioscience); PE anti-TIM-3 (344823; 1:20—#FAB2365P) (R&D systems). A fixable live/dead dye was used to distinguish viable cells (Biolegend). Cell surface staining was performed in FACS buffer (PBS, supplemented with 1% FBS and 0.01% NaN₃). Intracellular staining was performed using the Fix/Perm kit from eBioscience according to the manufacturer's instructions. To analyze cytokine production by PBMCs and TIL ex vivo, cells were stimulated for 5 h with Phorbol 12-Myristate 13-Acetate (PMA) (0.2 μM) and ionomycin (1 μg/ml), with BFA (10 μg/ml) present for the last 2.5 h. Intracellular cytokine staining was performed using the CytoFix/CytoPerm kit from BD Biosciences according to the manufacturer's instructions. Stained cells were acquired on a LSRII and LSRFortessa flow cytometer, or the FACS AriaII (all BD Biosciences), for cell sorting. Data were analyzed with FlowJo software (Treestar).

**Cell sorting and T-cell expansion.** Cryopreserved PBMC and TIL were thawed and enriched for T lymphocytes using the T-cell enrichment kit from Stemcell. For TIL enrichment, EpCAM beads (StemCell) were added to the cocktail. The enriched fractions were then labeled and populations of interest were purified after cell sorting to 99% purity on a FACSAria II. Briefly, naive CD8 T cells were sorted as CD8⁺CD4⁻CD45RA⁺CCR7⁺ cells and memory CD8 T cells were sorted as CD8⁺CD4⁻CD45RA⁻CCR7⁺/⁻ (total memory). CD8 subsets from TIL were sorted as CD3⁺CD4⁻CD8⁺CD45RA⁻CCR7⁺/⁻CD39⁻CD103⁻ (DN), CD3⁺CD4⁻CD8⁺CD45RA⁻CCR7⁺/⁻CD39⁻CD103⁺ (SP), and CD3⁺CD4⁻CD8⁺CD45RA⁻CCR7⁺/⁻CD39⁺CD103⁺ (DP). For TCR sequencing analysis, cell pellets were frozen until further processing.

For expansion of DN, SP, and DP CD8 TILs as well as naive and memory CD8 from PBMC, T cells were sorted and cultured in complete RPMI-1640, supplemented with 2 mM glutamine, 1% (vol/vol) nonessential amino acids, 1% (vol/vol) sodium pyruvate, penicillin (50 U/ml), streptomycin (50 μg/ml), and 10% fetal bovine serum (Hyclone). Of note, for functional assays and expansion, no CD3 antibody was used for cell sorting. Sorted T cells were stimulated polyclonally with 1 μg/ml Phytohemagglutinin (PHA) (Sigma) in the presence of irradiated (4000 rad) allogeneic feeder cells (PBMC; 2 × 10⁵ cells per well) and 10 ng/ml of interleukin (IL)-15 (Biolegend) in a 96-well round-bottom plate (Corning/Costar). T-cell lines were maintained in complete medium with IL-15 until analysis.

**Microarray data acquisition.** Samples for microarray were sorted as indicated above. Sorted cells were lyzed and RNA was purified using Direct-zol RNA MiniPrep kit (Zymo Research). RNA was then reverse transcribed to complementary DNA (cDNA) and amplified. Amplified cDNA was hybridized on a Affymetrix Prime-View gene chip.

**Microarray data analysis.** Image processing and expression analysis were performed using Affymetrix GeneChip Command Console (AGCC) v.3.1.1 and Affymetrix Expression Console v.1.1 software, respectively. Image processing of sample DAT files to generate sample CEL files was completed in Affymetrix Command Console software. Next, CEL were uploaded to Expression Console and were normalized using the RMA normalization and summarization algorithm. To determine differentially expressed genes between DP and DN CD8 TIL cells, we used the Limma package[45] lmFit to assess significance using half of the probes with the greatest overall variance (24,361). p-values were adjusted using the Benjamini–Hochberg method and probes with adjusted p-values <0.05 were considered significant. This resulted in 303 probes, mapping to 219 unique genes. A heatmap was constructed using expression values for the unique genes normalized to the mean across all cells. PCA was performed on the significantly differential

probes using the princomp function in the R statistical language (R Core Team (2016). R: A language and environment for statistical computing. R Foundation for Statistical Computing, Vienna, Austria).

The functional enrichment of DP vs. DN genes and DP vs. SP genes was analyzed by ranking all genes based on fold-change difference in expression, then applying GSEA[46], to test enrichment of several previously published T-cell gene signatures. The T$_{RM}$ signature was constructed from several studies[47–49] as the union set of up or downregulated genes from each. The false-discovery rate-adjusted p-values are reported (q value), with q values <0.05 considered significant.

**In vitro T-cell activation.** Naive CD8 T-cell subsets were isolated by magnetic CD8 T-cell enrichment (Stemcell), labeled with antibodies against CD4, CD8, CD45RA, and CCR7, and sorted as indicated above. 1 × 10⁵ naive T cells were cultured with anti-CD3/CD28 Dynabeads (Life Technologies) at a bead:T-cell ratio of 1:2 in the presence or absence of 2 ng/ml TGFβ-1 (R&D). After 24 h, beads were removed by magnetic capture for half of the experiment. Expression of activation and differentiation markers was assessed at days 1, 2, 3, 4, 7, and 9.

**DNA preparation and TCRβ deep sequencing.** Deep sequencing of the variable V–J or V–D–J regions of TCRβ genes was performed on genomic DNA of sorted T-cell populations. DNA was extracted from circulating and tumor-resident CD8 T-cell subsets ranging from 1 × 10⁴–1 × 10⁵ cells (DNeasy Blood and Tissue Kit, Qiagen). The TCRβ CDR3 regions were sequenced and mapped (ImmunoSEQ, Adaptive Biotech). Coverage per sample was >10×. Only data from productive rearrangements were extracted from the ImmunoSEQ Analyzer platform for further analysis. Clonality of the different T-cell subsets was assessed by nucleotide sequence comparison of the 500 most abundant clones in each subset.

To compare the TCR Vβ overlap (or similarity) of two given populations, we used the Morisita's overlap and the Jaccard index. The Morisita–Horn similarity index accounts for both, the number of common clonotypes and the distribution of clonotype sizes, and is most sensitive to the clone sizes of the dominant clonotypes[50]. The Jaccard index accounts for the proportion of shared clonotypes between samples.

**Assessment of target cell recognition.** Upregulation of 4-1BB as well as release of IFN-γ were used as measures to assess recognition of tumor cells by expanded autologous CD8 T cells. The co-culture experiment was performed 17–20 days after expansion. Expanded CD8 T cells (1 × 10⁵) were then cultured either alone or with increasing numbers of autologous tumor cells (ranging from 781 to 100,000 cells per well). In some conditions, tumor cells were pre-incubated with 50 μg/ml of anti-MHC-class I blocking antibody (Biolegend, clone W6/32—#311428) for 1 h prior to adding T cells. To confirm tumor reactivity, expanded CD8 T cells were cultured with allogeneic tumor cells. As positive control, Nunc Maxisorp plates were coated with 1 μg/ml anti-CD3 antibody (OKT3) and T cells were added. All conditions were plated in triplicate. After 20 h, supernatants were harvested and analyzed by enzyme-linked immunosorbent assay (ELISA). Cells were collected and labeled with a viability dye, followed by CD39, CD103, CD25, and 4-1BB cell surface staining. Cells were analyzed by flow cytometry. For the ELISA, the manufacturer's protocol (BD Biosciences) was followed and secretion of IFN-γ was analyzed.

**Live target cell killing assay.** T-cell-mediated target cell killing assays were performed in the Incucyte Zoom System housed inside a cell incubator at 37 °C/5% CO₂, based on the manufacturer's protocol (Essen Bioscience). To assess T-cell killing of autologous tumor cells, 5000–10,000 tumor cells (autologous and allogeneic tumor cell lines) were seeded in triplicate in a 96-well flat-bottom plate to reach 10% confluence. Expanded autologous T-cell subsets were counted and 1 × 10⁵ T cells were cultured with and without autologous and allogeneic tumor cells in medium without exogenous IL-15 (T cell:tumor cell ratio of 10:1). In some wells, anti-MHC-class I antibody (Biolegend, clone W6/32) was added. In all conditions, NucView 488 Caspase 3/7 substrate (Essen Bioscience) was added to monitor active caspase 3/7. Plates were incubated for 24 h at 37 °C and four images were captured from three experimental replicates every hour using a ×10 objective lens to visualize T-cell killing and caspase 3/7 activity (green fluorescence). Green channel acquisition time was 400 ms. For phase contrast, cell segmentation was achieved by applying a mask to exclude the smaller T cells. An area filter was applied to exclude objects below 1000 μm². Green background noise was subtracted with the Top-Hat method of background non-uniformity correction with a radius of 20 μm and a threshold of two green corrected units. Fluorescence signal was quantified after applying the mask to the experiment. Amount of T-cell killing/apoptosis was calculated by the Zoom software provided (Essen Bioscience).

**IFN-γ Enzyme-Linked ImmunoSpot (ELISPOT) assay.** Briefly, ELIIP plates (Millipore, MAIPSWU10) were pretreated with 50 μl per well of 70% ethanol for 2 min, washed 3× with PBS, and then coated with 50 μl of 10 μg/ml IFN-γ capture antibody (Mabtech, clone: 1-D1K), and incubated overnight in the fridge. For OKT3 controls, wells were coated with a mixture of IFN-γ capture antibody (10 μg/ml) and OKT3 (1 μg/ml). Prior to co-culture, the plates were washed 1× with PBS, followed by blocking with complete RPMI media for at least 1 h at room temperature. After 24 h of co-culture, the ELISPOT plates were washed 6× with PBS +

0.05% Tween-20 (PBS-T), and then incubated for 2 h at room temperature with 100 μl per well of a 0.22 μm-filtered 1 μg/ml biotinylated anti-human IFN-γ detection antibody solution (Mabtech, clone: 7-B6-1, diluent consisted of 1× PBS supplemented with 0.5% FBS). The plate was then washed 3× with PBS-T, followed by 1 h incubation with 100 μl per well of streptavidin-ALP (Mabtech, diluted 1:3000 with above diluent). The plate was washed 5× with PBS followed by development with 100 μl per well of 0.45 μm-filtered BCIP/NBT substrate solution (KPL, Inc.). The reaction was stopped by rinsing thoroughly with cold tap water. ELISPOT plates were scanned and counted using an ImmunoSpot plate reader and associated software (Cellular Technology Limited).

**Survival data and analysis**. For the OS analysis, the survival time was calculated from the time of surgery until last contact or death. The censoring proportion of survival data was 47 of 62 patients (for details, see Supplementary Table 5). The frequency of DP CD8 T cells among total CD8 TILs in all 62 HNSCC tumors was evaluated at time of surgery. We compared OS between DP low and DP high using the mean frequency of DP CD8 TILs calculated from all 62 patients as a cut point (mean = 32.9%). This cut point was not only clinically important but also best discriminate patient survival, as confirmed using the score (or log-rank) statistics in the R function cutp() from the package "survMisc" (Chris Dardis (2016). survMisc: Miscellaneous Functions for Survival Data. R package version 0.5.4; https://cran.r-project.org/web/packages/survMisc/index.html). Patient characteristics between groups were evaluated using Fisher exact tests for categorical, and *t* test or non-parametric Kruskal–Wallis rank sum tests for continuous variables. Univariate survival analysis was performed using the Kaplan–Meier method and *p*-values were determined using log-rank test. Multivariable survival analysis was performed using Cox-proportional model with Firth's penalized likelihood (Heinze and Ploner (2016). coxphf: Cox regression with Firth's penalized likelihood. R package version 1.12; https://CRAN.R-project.org/package=coxphf), which has been shown as a good alternative to Cox's regression model with a more precise estimation of parameters for small samples with substantial censoring of survival times[51]. It produces finite parameter estimates using penalized maximum likelihood estimation for monotone likelihood that primarily occurs in small samples with high percentage of censoring observed in a Cox model. Therefore, we adopted this penalization approach for the multivariable survival analysis in this study. In order to fully capture the effect of DP CD8 on OS, we also performed the survival analysis taking DP CD8 as continuous variable with log(2) transformed without dichotomizing them. All statistical tests were two-sided tests. The statistical analyses were performed using R statistical software, version 3.3.2.

**Survival data and analysis TCGA**. The data used was downloaded from the TCGA repository. All available mRNA expression data was downloaded from the following projects: TCGA-HNSC (head and neck squamous cell carcinoma), TCGA-LUAD (lung adenocarcinoma), and TCGA-LUSC (lung squamous cell carcinoma). The clinical data for the same projects was downloaded in R using the RTCGA package. We analyzed patient survival according to the expression of the transcripts for *ENTPD1* (CD39) and *ITGAE* (CD103). To determine the ideal point to divide groups, the R package 'survminer' was used. The surv_cutpoint() function, which uses the maximally selected rank statistics from the 'maxstat' R package, was used to split each of the genes into high- or low-expresser groups. Following this analysis, the CD39hiCD103hi and CD39loCD103lo groups were established and isolated. The survival and ggplot2 R packages were used to create Kaplan–Meier plots for each of the selected types of cancer including *p*-values determined by log-rank test.

**Statistical analysis**. Statistical significance between groups was determined by one-way analysis of variance analysis with Tukey correction, using GraphPad Prism 6 software (GraphPad, San Diego, CA).

**Data availability**. All data are available from the corresponding authors upon reasonable request. Gene-expression raw data that support the findings of this study have been deposited in Gene Expression Omnibus under the accession number GSE114944. TCR sequencing data have been deposited in the ImmuneACCESS database under the accession code DOI:10.21417/B7ZW6H (http://clients.adaptivebiotech.com/pub/Duhen-2018-natcomms).

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

## Acknowledgements

We thank R. Tamakawa and I.F. Hilgart Martiszus for assistance with consenting patients for this study; K.A. Bright and J. Birch for processing patient samples; P.H. Newell, P. Hansen, and P. Tseng for providing tumor samples; E. Tran and W.J. Urba for reviewing the manuscript and discussions. Microarray assays were performed in the OHSU Gene Profiling Shared Resource. Supported by the National Cancer Institute (CA102577 for A.D.W. and CPTAC award U24CA160019 for J.E.M.) and the PNNL Technology Assistance Program (J.E.M.).

## Author contributions

T.D., R.D., R.M. and A.D.W. designed the experiments; T.D., R.D. and R.M. performed the experiments; J.M. performed the survival analysis on the TCGA data set and the viral-specific analysis using the VDJ database; C.P.G. performed ELISA assays; T.C.B. performed initial experiment on tumor reactivity; T.D., R.D. and R.M. analyzed the data; J.E.M. analyzed the microarray data; S.-C.C. analyzed patient survival under the super-vision of G.G.; R.L. and R.B.B. assisted in patient recruitment, obtaining consent, and sample collection; T.M. and B.A.F. provided tumor cell lines and tumor digests for tumor-reactivity experiments; N.F.d.M. provided tumor samples from colon cancer patients; T.D., R.D. and A.D.W. wrote the manuscript; A.D.W. initiated the research program with R.M. and supervised the study.
