## [Peer Review File · Nature Communications]

Editorial Note: This manuscript has been previously reviewed at another journal that is not operating a transparent peer review scheme. This document only contains reviewer comments and rebuttal letters for versions considered at Nature Communications .

Reviewers' comments:

Reviewer #1 (Cancer, T response)(Remarks to the Author):

This manuscript has been substantially improved, and most of the major concerns have been addressed with additional data and/or clarification. However, there are still some remaining issues:

Major:

1. Lines 229-230. It's somewhat surprising that the percentage (rather than absolute number) of DP CD8+ TIL relative to total CD8+ TIL is associated with OS. In theory, this means that even tumors with a low CD8 infiltrate could score highly as long as they had an above-threshold percentage of DP CD8+ TIL. Is that the case? It would be helpful to clarify this, as most TIL-based prognostic studies utilize the density of TIL relative to tumor area.

2. Line 241. Why was lung cancer chosen for this analysis, given that this cancer type is not evaluated elsewhere in the paper? Were other cancer types in TCGA also evaluated? If so, all results should be reported to avoid publication bias.

3. Discussion. The authors imply that CD103+CD39- T cells are T_{rm} whereas CD103+CD39+ T cells are bona fide TIL. Is this their conclusion? If so, it would be helpful to state it more explicitly, even if it's in the form of a hypothesis.

4. Supp Fig 6. It would be helpful to show the equivalent data for the other tumor types analyzed, such as ovarian cancer.

Minor:

1. Line 166. "indicate" should be changed to "suggests" since there is no direct evidence that T cells are expanding at the tumor site.

2. Line 299. Perhaps "skewed" is a more accurate term than "oligoclonal" to describe the TCR-seq results?

Reviewer #2 (Lung cancer, ICB)(Remarks to the Author):

I think that the manuscript has been greatly enhanced by the responses to the comments. I believe that the additional patients have certainly added to the conclusions, and think that the data is now easier to interpret.

Reviewer #3 (Cancer, T exhaustion)(Remarks to the Author):

In their revision, Duhon et al. have significantly improved their manuscript, addressing almost all of our concerns with their additional analyses. Our major concern - that their work was incremental compared with using CD103+ as a marker alone (as has been previously published) -

has been adequately addressed. Their functional data has been validated in 4 additional patient samples and they added many more patients to their flow characterization. TCGA analysis strengthens their survival correlation data, although we still have a few reservations about the way that they present this data. We understand that they are working with a small sample size (n=62) of HNSCC patients, however it is clear from Supp Table 2 that there is potential of confounding of DP low vs high status with stage of disease, which could affect their OS analysis. They attempt to address this by performing univariable and multivariable analysis (Supp Table 3), however, it does not make biologic sense that stage does not correlate with survival. They further restrict their analysis to HPV-negative patients, which is logical given the differential survival between HPV+ and HPV- patients, but then fail to show the entire univariable and multivariable table for this subanalysis (Supp Table 4). Providing more detail of this analysis in the supplemental tables, and rewriting the results or discussion to acknowledge the confounding variables and their multivariable analysis to account for them, would significantly strengthen their manuscript.

Of note, the reference to Supp Figure 8 on line 215 is incorrect, and should be changed to Supp Figure 7

Reviewer #4 (Biostat, cancer trial)(Remarks to the Author):

Summary

This submission is a revision of the transferred manuscript from Nature Medicine. The major statistical concern from the previous reviewers was small sample sizes. In this revision, authors seemed to have increased sample sizes to a reasonable level that might provide statistically stable results. In terms of statistical methods used, the log-rank test and Cox proportional hazards model, allowing for small sample analysis, were used to compare time-to-event distributions with and without adjusting for confounding factors, respectively. Morisita's index was used to compare the overlap among samples, and the Jaccard index has been added to this revision to compare the similarity and diversity of sample sets. For the analysis of gene expression data to find probes that differentiate DP and DN CD8 TIL cells, the Benjamini-Hochberg method was used to control the false discovery rate (FDR), and the principal component (PCA) was used to identify the main factors that would explain most variations in the significantly differential probes. The statistical methods used are all appropriate and the analyses seem to be reasonable, but here are some suggestions to improve the clarity:

1. In the section of Survival data and analysis, looking at various potential cutoff points to determine an optimal one that provide the largest difference would lead to a multiple testing issue. Therefore, the final test based on the selected optimal cutoff point needs to be carefully performed, e.g. see Section 8.6 in Survival Analysis: Techniques for Censored and Truncated Data by Klein and Moeschberger.
2. In the Section of Survival data and analysis, the censoring proportion for survival data should be mentioned.
3. In the Section of Survival data and analysis, it should be clearly mentioned whether the statistical tests were one-sided or two-sided.
4. In the Supplementary Materials, Tables 5 and 6 contained the results from Fisher's exact test and Kruskal-Wallis test, which can be also mentioned in the Section of Survival data and analysis.
5. Supplementary tables 5 and 6 may need to be reformatted.
6. Line 683, pg 30; I would use "the mean" instead of "mean".

7. Line 696, pg 30; It seems that "primary" should be "primarily".

8. Line 706, pg 31; Authors indicate that they used the R package "survminer" to determine the ideal point, but the package is mainly for high-quality graphics. Does this mean that it was done graphically? Please clarify.

Point-by-point response to reviewer's comments

#NCOMMS-18-09195A

“Co-Expression of CD39 and CD103 identifies tumor-reactive CD8 TIL in human solid tumors”

Reviewer #1 (Cancer, T response)(Remarks to the Author):

This manuscript has been substantially improved, and most of the major concerns have been addressed with additional data and/or clarification. However, there are still some remaining issues:

Major:

1. Lines 229-230. It's somewhat surprising that the percentage (rather than absolute number) of DP CD8+ TIL relative to total CD8+ TIL is associated with OS. In theory, this means that even tumors with a low CD8 infiltrate could score highly as long as they had an above-threshold percentage of DP CD8+ TIL. Is that the case? It would be helpful to clarify this, as most TIL-based prognostic studies utilize the density of TIL relative to tumor area.

Based on our results, we believe that the percentage of DP cells within the CD8 TIL has a strong correlation with OS in our HNSCC cohort. Since these experiments were performed by flow cytometry it was very hard to quantitate exact numbers of cells because some of the tumor samples were extremely small and for most of the samples we used several antibody panels for the analysis. We also found that there was no correlation between the frequency of CD8 TIL and the percentage of DP CD8 (**Supplementary Fig. 8a**). Hence, some patient samples had low frequencies of CD8s but a high proportion of them were DP and those patients tended to have a survival advantage. We are working on an IHC based assay to detect CD39 and CD103 on CD8 TIL (not perfected quite yet) and with this assay we can take into account the absolute number of CD8 to address this point in future studies. It should also be noted that we found a similar correlation with CD39 and CD103 mRNA expression in the TCGA database with HNSCC (**Supplementary Fig. 8b**). However, this was performed with total RNA and therefore we were not directly assessing the expression of these two markers on CD8 T cells.

2. Line 241. Why was lung cancer chosen for this analysis, given that this cancer type is not evaluated elsewhere in the paper? Were other cancer types in TCGA also evaluated? If so, all results should be reported to avoid publication bias.

We chose to study the survival analysis in lung cancer patients for two reasons:

1) We showed in our manuscript in Fig. 1d that DP CD8 TIL were present at fairly high frequencies in lung cancer patients, and 2) Two previous studies (Djenidi F et al, J

Immunol 2015 and Ganesan AP et al, Nat Immunol 2017) have shown that CD8+CD103+ T cells were associated with better survival in lung cancer patients. Therefore, we were interested to assess whether increased RNA for CD39 and CD103 could also predict greater survival compared to CD103 alone. Hence, we focused our TCGA analyses in HNSCC and lung cancer patient data sets. We plan to assess other tumor types in future studies for a survival advantage.

3. Discussion. The authors imply that CD103+CD39- T cells are Trm whereas CD103+CD39+ T cells are bona fide TIL. Is this their conclusion? If so, it would be helpful to state it more explicitly, even if it's in the form of a hypothesis.

It is not clear whether the CD103+CD39- CD8 TIL are Trm as they seem to have an intermediate phenotype in terms of the Trm gene signature (**Fig. 2e**). Since the CD103+CD39- CD8 T cells express CD103 and CD69 (see **Fig. 3a**) it would tend to suggest that they could be Trm cells based on previous literature, but we do not have any further data to support that they are indeed this lineage of cells. Based on our TCRseq analysis, there is very little overlap between the two CD103 subsets (**Fig 5b and d**), hence the CD103+CD39- CD8 TIL could be recently activated T cells in the periphery (e.g. pathogen specific T cells) that are recruited to a TGF-b-rich microenvironment explaining their up-regulation of CD103 but little to no tumor reactivity.

4. Supp Fig 6. It would be helpful to show the equivalent data for the other tumor types analyzed, such as ovarian cancer.

We agree with the reviewer that it would be interesting to determine if similar results would be observed in other tumor types. Unfortunately, there is a limited number of patient samples from which we were able to generate autologous cell lines for this study. We were able to generate tumor cell lines from six patient samples and saw consistent results across all six samples, therefore we believe that this data set will most likely be applicable to the other tumor types studied in this manuscript.

Minor:

1. Line 163. "indicate" should be changed to "suggests" since there is no direct evidence that T cells are expanding at the tumor site.

As recommended by the reviewer, line 163 “indicate” was changed to “suggest”.

2. Line 306. Perhaps "skewed" is a more accurate term than "oligoclonal" to describe the TCR-seq results?

As recommended by the reviewer, line 306 “oligoclonal” was changed to “skewed”.

Reviewer #2 (Lung cancer, ICB)(Remarks to the Author):

I think that the manuscript has been greatly enhanced by the responses to the comments. I

believe that the additional patients have certainly added to the conclusions, and think that the data is now easier to interpret.

Reviewer #3 (Cancer, T exhaustion)(Remarks to the Author):

In their revision, Duhon et al. have significantly improved their manuscript, addressing almost all of our concerns with their additional analyses. Our major concern - that their work was incremental compared with using CD103+ as a marker alone (as has been previously published) - has been adequately addressed. Their functional data has been validated in 4 additional patient samples and they added many more patients to their flow characterization. TCGA analysis strengthens their survival correlation data, although we still have a few reservations about the way that they present this data. We understand that they are working with a small sample size (n=62) of HNSCC patients, however it is clear from Supp Table 2 that there is potential of confounding of DP low vs high status with stage of disease, which could affect their OS analysis. They attempt to address this by performing univariable and multivariable analysis (Supp Table 3), however, it does not make biologic sense that stage does not correlate with survival. They further restrict their analysis to HPV-negative patients, which is logical given the differential survival between HPV+ and HPV- patients, but then fail to show the entire univariable and multivariable table for this subanalysis (Supp Table 4). Providing more detail of this analysis in the supplemental tables, and rewriting the results or discussion to acknowledge the confounding variables and their multivariable analysis to account for them, would significantly strengthen their manuscript.

We agree with the reviewer that it is surprising that pathologic stage does not correlate with survival. We believe that it may be due to small sample size and small number of events [total n=62 with 15 events (i.e., deaths)], and hence insufficient power to detect a significant difference within each stage of disease. Also, the majority of the patients had stage IV disease making it difficult to calculate significance using stage as a parameter. We added sentences in the results section regarding this point and to acknowledge the confounding variables and our multivariable analysis to account for them (Lines 225 to 229 and 235 to 245).

We apologize for not showing the entire univariate and multivariable table for the analysis of the HPV-negative cohort of patients. We have now replaced the Supplementary Table 4 with a new version that contains the entire analysis.

Of note, the reference to Supp Figure 8 on line 211 is incorrect, and should be changed to Supp Figure 7

As recommended by the reviewer, the reference to Supplementary Figure 8 in line 211 has been changed to Supplementary Figure 7.

Reviewer #4 (Biostat, cancer trial)(Remarks to the Author):

Summary

This submission is a revision of the transferred manuscript from Nature Medicine. The major statistical concern from the previous reviewers was small sample sizes. In this revision, authors seemed to have increased sample sizes to a reasonable level that might provide statistically stable results. In terms of statistical methods used, the log-rank test and Cox proportional hazards model, allowing for small sample analysis, were used to compare time-to-event distributions with and without adjusting for confounding factors, respectively. Morisita's index was used to compare the overlap among samples, and the Jaccard index has been added to this revision to compare the similarity and diversity of sample sets. For the analysis of gene expression data to find probes that differentiate DP and DN CD8 TIL cells, the Benjamini-Hochberg method was used to control the false discovery rate (FDR), and the principal component (PCA) was used to identify the main factors that would explain most variations in the significantly differential probes. The statistical methods used are all appropriate and the analyses seem to be reasonable, but here are some suggestions to improve the clarity:

1. In the section of Survival data and analysis, looking at various potential cutoff points to determine an optimal one that provide the largest difference would lead to a multiple testing issue. Therefore, the final test based on the selected optimal cutoff point needs to be carefully performed, e.g. see Section 8.6 in Survival Analysis: Techniques for Censored and Truncated Data by Klein and Moeschberger.

We did not perform multiple tests to select an optimal cut-point. We used the mean throughout the study. Other cutoffs were explored to check if the conclusion changed at these different measurements. In order to fully capture the effect of DP CD8 TIL on overall survival, we also performed the survival analysis taking DP CD8 TIL as a continuous variable with log(2) transformed without dichotomizing them (Supplementary table 3 and 4).

2. In the Section of Survival data and analysis, the censoring proportion for survival data should be mentioned.

The censoring proportion for survival data is 47 over 62 patients (details are in Supplementary table 5). The censoring proportion has been added to the Section of Survival data and analysis (Lines 534 - 535).

3. In the Section of Survival data and analysis, it should be clearly mentioned whether the statistical tests were one-sided or two-sided.

Two-sided tests were performed. This has been added to the Section of Survival data and analysis (Lines 555 - 556).

4. In the Supplementary Materials, Tables 5 and 6 contained the results from Fisher's

exact test and Kruskal-Wallis test, which can be also mentioned in the Section of Survival data and analysis.

In the Supplementary section, Tables 5 and 6 contain the results from log-rank test. Fisher's exact test and Kruskal-Wallis test were used in Table 2. The information regarding all the tests discussed above is now included in the Section of Survival data and analysis.

5. Supplementary tables 5 and 6 may need to be reformatted.

As recommended by the reviewer, both tables have been reformatted.

6. Line 537, pg 25; I would use "the mean" instead of "mean".

As recommended by the reviewer, line 537 "mean" was replaced by "the mean".

7. Line 551, pg 25; It seems that "primary" should be "primarily".

As recommended by the reviewer, line 551 "primary" was replaced by "primarily".

8. Line 566, pg 26; Authors indicate that they used the R package "survminer" to determine the ideal point, but the package is mainly for high-quality graphics. Does this mean that it was done graphically? Please clarify.

We apologize for the confusion. We modified the sentence as indicated below:

"To determine the ideal point to divide groups the R package 'survminer' was used. The `surv_cutpoint()` function, which uses the maximally selected rank statistics from the 'maxstat' R package, was used to split each of the genes into high or low expresser groups."

REVIEWERS' COMMENTS:

Reviewer #1 (Remarks to the Author):

The issues I raised have been addressed, and the manuscript is significantly improved. There are no remaining issues.

Reviewer #3 (Remarks to the Author):

The comments have been addressed satisfactorily.

Reviewer #4 (Remarks to the Author):

All the responses are acceptable except the first one, where this reviewer was NOT asking if a multiple testing was used to select the optimal cutoff point. The issue was rather if the optimal cutoff point was selected by exploring multiple points and used for a hypothesis testing, then the type I error could be inflated. Therefore, if you used the SELECTED optimal cutoff point for a hypothesis testing, the test should be adjusted accordingly to control the Type I error. If you didn't use the SELECTED optimal cutoff point for a hypothesis testing, this issue is not relevant. Please clarify.

Point-by-point response to reviewer's comments

#NCOMMS-18-09195A

“Co-expression of CD39 and CD103 identifies tumor-reactive CD8 T cells in human solid tumors”

##Referee4 comments##

All the responses are acceptable except the first one, where this reviewer was NOT asking if a multiple testing was used to select the optimal cutoff point. The issue was rather if the optimal cutoff point was selected by exploring multiple points and used for a hypothesis testing, then the type I error could be inflated. Therefore, if you used the SELECTED optimal cutoff point for a hypothesis testing, the test should be adjusted accordingly to control the Type I error. If you didn't use the SELECTED optimal cutoff point for a hypothesis testing, this issue is not relevant. Please clarify.

We apologize for the confusion. We did not use the SELECTED optimal cutoff point for a hypothesis testing in our study. We hope that our answer clarifies the point that was raised.